# Current Positioning against Severe Infections Due to *Klebsiella pneumoniae* in Hospitalized Adults

**DOI:** 10.3390/antibiotics11091160

**Published:** 2022-08-27

**Authors:** Pablo Vidal-Cortés, Ignacio Martin-Loeches, Alejandro Rodríguez, Germán Bou, Rafael Cantón, Emili Diaz, Carmen De la Fuente, Julián Torre-Cisneros, Francisco Xavier Nuvials, Miguel Salavert, Gerardo Aguilar, Mercedes Nieto, Paula Ramírez, Marcio Borges, Cruz Soriano, Ricard Ferrer, Emilio Maseda, Rafael Zaragoza

**Affiliations:** 1ICU, Hospital Universitario Ourense, 32005 Ourense, Spain; 2ICU, Trinity Centre for Health Science HRB-Welcome Trust, St. James’s Hospital Dublin, D08 NHY1 Dublin, Ireland; 3ICU, Hospital Universitari Joan XXIII, 43005 Tarragona, Spain; 4Institut d’Investigació Sanitària Pere Virgil, 43007 Tarragona, Spain; 5Departament Medicina I Ciruurgia, Universitat Rovira i Virgili, 43003 Tarragona, Spain; 6Centro de Investigación en Red de Enfermedades Respiratorias (CIBERES), Instituto de Salud Carlos III, 28029 Madrid, Spain; 7Microbiology Department, Complejo Hospitalario Universitario A Coruña, 15006 A Coruña, Spain; 8Centro de Investigación en Red de Enfermedades Infecciosas (CIBERINFEC), Instituto de Salud Carlos III, 28029 Madrid, Spain; 9Microbiology Department, Hospital Universitario Ramón y Cajal, Instituto Ramón y Cajal de Investigación Sanitaria (IRYCIS), 28034 Madrid, Spain; 10Critical Care Department, Corporació Sanitària Parc Taulí, 08208 Sabadell, Spain; 11Department of Medicine, Universitat Autonoma de Barcelona (UAB), 08193 Barcelona, Spain; 12ICU, Hospital Universitario Reina Sofía, 14004 Córdoba, Spain; 13Infectious Diseases Service, Hospital Universitario Reina Sofía, 14004 Córdoba, Spain; 14Instituto Maimónides de Investigación Biomédica de Córdoba (IMIBIC), Universidad de Córdoba, 14004 Córdoba, Spain; 15ICU, Hospital Vall d’Hebrón, 08035 Barcelona, Spain; 16Infectious Diseases Department, Hospital Universitari I Politecnic La Fe, 46026 Valencia, Spain; 17SICU, Department of Anesthesiology and Critical Care, Hospital Clínico Universitario Valencia, 46014 Valencia, Spain; 18School of Medicine, Universitat de Valencia, 46010 Valencia, Spain; 19ICU, Hospital Clínico Universitario San Carlos, 28040 Madrid, Spain; 20ICU, Hospital Universitari I Politecnic La Fe, 46026 Valencia, Spain; 21ICU, Hospital Universitario Son Llázter, 07198 Palma de Mallorca, Spain; 22Fundación Micellium, 46183 Valencia, Spain; 23ICU, Hospital Universitario Ramón y Cajal, 28034 Madrid, Spain; 24SICU, Hospital Universitario La Paz, 28046 Madrid, Spain; 25ICU, Hospital Universitario Dr. Peset, 46017 Valencia, Spain

**Keywords:** carbapenemase-producing Enterobacterales (CPE), *Klebsiella pneumoniae*, KPC, VAP, nosocomial pneumonia, ceftazidime-avibactam, imipenem-relebactam, meropenem-vaborbactam, cefiderocol, ICU

## Abstract

Infections due to *Klebsiella pneumoniae* have been increasing in intensive care units (ICUs) in the last decade. Such infections pose a serious problem, especially when antimicrobial resistance is present. We created a task force of experts, including specialists in intensive care medicine, anaesthesia, microbiology and infectious diseases, selected on the basis of their varied experience in the field of nosocomial infections, who conducted a comprehensive review of the recently published literature on the management of carbapenemase-producing Enterobacterales (CPE) infections in the intensive care setting from 2012 to 2022 to summarize the best available treatment. The group established priorities regarding management, based on both the risk of developing infections caused by *K. pneumoniae* and the risk of poor outcome. Moreover, we reviewed and updated the most important clinical entities and the new antibiotic treatments recently developed. After analysis of the priorities outlined, this group of experts established a series of recommendations and designed a management algorithm.

## 1. Introduction

Infections caused by *Klebsiella pneumoniae* have increased in hospitalization wards, especially critical care settings, in the last decade, and they pose a severe problem when antimicrobial resistance is present. This point-of-view article summarizes the recently published literature on the management of carbapenemase-producing Enterobacterales (CPE) infections in critically ill patients to determine the best available treatments for patients while trying to avoid inappropriate antimicrobial therapy.

## 2. Current Epidemiology, Pathogeny and Antimicrobial Resistance of *Klebsiella pneumoniae* Infections in the ICU

*K. pneumoniae* is one of the microorganisms most commonly implicated in healthcare-associated infections (HAIs), accounting for 2% to 5% of hospital infections between 2011 and 2014 in Spain and nearer to 8% in more recent years, and is especially frequent among infections that affect the urinary and respiratory tracts [1]. A recent one-day prevalence study conducted worldwide found that *Klebsiella pneumonia* was the most frequently isolated pathogen in ICUs. Compared with other pathogens, *Klebsiella* causes 5% to 10% of infections in the ICU, and some high resistance percentages have been observed, sometimes close to 50%, against third-generation cephalosporins, penicillins associated with β-lactamase inhibitors and quinolones, as well as growing resistance to carbapenems [2].

At least four components are involved in the pathogenicity shown by *Klebsiella* isolates: the existence of fimbriae or the production of adhesins, the expression of siderophores, the existence of a lipopolysaccharide that protects this microorganism from patient immune responses and the presence of a polysaccharide capsule [3,4,5].

Resistance to carbapenems (imipenem, meropenem and ertapenem) in *K. pneumoniae* is mediated by two major mechanisms. First, through the production of b-lactamases with the ability to hydrolyze cephalosporins, such as ESBL (e.g., CTX-M-type) and AmpC cephalosporinase (e.g., DHA-1 or CMY-2), in combination with decreased membrane permeability in the cell wall [6]. The second mechanism is mediated by the production of b-lactamases capable of hydrolyzing carbapenems, namely, carbapenemases. According to the Ambler classification, carbapenemases can be classified as serine b-lactamases, including class A (*K. pneumoniae* carbapenemase (KPC), as well as BKC and SME, among others) and class D (OXA-48-like) enzymes, as well as class B or metallo-b-lactamases (MBLs), including Verone-imipenemase (VIM), imipenemase (IMP) and New Delhi metallo-b-lactamases (NDMs) [7,8,9]. These types of enzymes have structural and biochemical differences, as well as different mechanisms of action. In this regard, MBLs require zinc for the hydrolysis of b-lactam antibiotics, and their activity can be inhibited by ethylenediamine tetra-acetic acid (EDTA) or dipicolinic acid as chelating agents [10], whereas most serine b-lactamases are inhibited by recently released b-lactamase inhibitors, such as avibactam, vaborbactam and relebactam [9,10].

KPC-producing *K. pneumoniae* isolates were first detected in the United States in 1996. The presence of these bacteria has subsequently been revealed in many other countries, including China, Italy and several South American countries [11]. The genes encoding KPC carbapenemases in this pathogen and other *Enterobacterales* can be located on mobile genetic elements, such as conjugative plasmids (e.g., IncFII, IncL/M and IncA/C), and in proximity to Tn4401 transposons and non-Tn4401 elements. However, transposon Tn4401 has been revealed to be the main genetic structure facilitating the dissemination of *bla* _KPC_-like genes to different plasmid scaffolds [7,10].

In regard to class D β-lactamases, only OXA-48 and variants such as OXA-163, -181, -232 and -244 have been reported in *K. pneumoniae*. The *bla*_OXA-48_ gene is located in the Tn1999 composite transposon [12]. The most common plasmids that harbour Tn1999 variants associated with *bla*_OXA-48_ belong to the IncL (IncL/M) replicon types [10]. pOXA-48 is highly transmissible, exhibiting a transfer frequency 50 times higher than those of the similar IncL plasmids with *bla*_NDM-1_. This is likely due to the insertion and disruption of Tn1999 variants into the *tir* regions of pOXA-48 (the *tir* gene encodes a protein that inhibits transfer) [13].

VIM-type and NDM-type MBLs are among the most prevalent MBLs found in this pathogen [11,14]. The NDM carbapenemase was reported from *K. pneumoniae* and *Escherichia coli* in 2009, similar to another kind of MBL [15]. The MBL genes (e.g., *bla*_IMP_
*,bla*_VIM_ and *bla*_NDM_) are found on different broad-host-range plasmid types (e.g., IncA/C and IncN) with various genetic features [16]. The *bla*_IMP_ and *bla*_VIM_ genes are usually found in class-1 integron structures located within transposon structures that enhance their dissemination. However, *bla*_NDM_ genes are associated with mosaic genetic structures, including insertion sequences (e.g., ISAba1), but the exact mechanism leading to their acquisition on plasmid scaffolds remains unknown [17]. *bla*_NDM_ has been found in a variety of genetic contexts, suggesting that multiple mechanisms are involved in the mobilization of *bla*_NDM_ [10].

Carbapenemases have variable and different hydrolytic activities, with MBLs and KPC enzymes being more efficient than OXA-48-like enzymes [10]. However, high-level carbapenem resistance among *K. pneumoniae* isolates with carbapenemases requires additional permeability deficiencies, regardless of the type of carbapenemase produced [18]. To hallmark this, isolates of *K. pneumoniae* with all kinds of carbapenemases showing low carbapenem MICs have been identified. This might account for the initial successful spread of *K. pneumoniae* with *bla*_KPC_ in the US during the 1990s and the initial spread of *K. pneumoniae* with *bla*_VIM_ in Greece, where some isolates with VIM-type enzymes had imipenem MICs lower than 0.5 mg/L so they were susceptible according to clinical breakpoints and undetectable for diagnostic systems of that time [8].

However, multidrug-resistant (MDR) *K. pneumoniae* isolates recovered in ICUs are associated with high-risk clones (HiRiCs). These are defined as highly specialized genetic populations or subpopulations with enhanced abilities to colonize, spread and persist in particular niches after having acquired diverse adaptive traits that increase their epidemicity and/or pathogenic potential, including antibiotic resistance [19]. The relevance of HiRiCs in this species was first highlighted after their association with the spread of ESBL and, more recently, with carbapenemases. In most cases, the same clones that initially acquired ESBL-producing plasmids were those that subsequently incorporated carbapenemase genes, demonstrating the importance of the “local clonal pool” in the maintenance of resistant populations and the acquisition of new resistance mechanisms. As a consequence, it is very common for carbapenamase-producing *K. pneumoniae* isolates to also produce ESBL [20]. Different HiRiCs of *K. pneumoniae* have revealed this situation (Table 1).

The spread of other HiRiCs has also been reported, including ST11 and ST147, both associated with other b-lactamases transmitted by horizontal genetic processes, such as plasmid ampC enzymes (e.g., DHA-1) and carbapenemases (KPC, VIM, NDM and OXA-48), in numerous countries [21,22,23]. However, the importance of these clones became more apparent with the detection of KPC-type carbapenemases in *K. pneumoniae* isolates belonging to ST258. Phylogenetic studies revealed the emergence of these strains during the early 2000s, with rapid global spread, cross-country transmission and widespread endemicity, with multiple outbreaks [24]. ST258 consists of two distinct lineages (clades I and II). The most widely accepted hypothesis is that they arose as a result of a recombination process between ST11 and ST442 clones. In addition, it was possible to acquire promiscuous plasmids from the incompatibility F group harbouring the *bla*_KPC-2_ or *bla*_KPC-3_ genes 21 [10].

Other very close HiRiCs of ST11 and ST258 are ST340, ST437 and ST512, all of which are single locus variants of ST258, comprising the so-called CC258 [25]. In some institutions, isolates associated with this CC acquired resistance to colistin, caused outbreaks in ICUs and were maintained over the years in the same institution, revealing the tenacity of the HiRiCs in these settings [26,27,28].

More recently, a HiRiC that has attracted and required significant attention due to the limitation of therapeutic options was the ST307 clone [22]. Its importance has been confirmed in epidemiological monitoring studies by its global dissemination and data obtained in population structure studies using next-generation sequencing techniques. As with other HiRiCs, phylogenetic studies suggested the emergence of this clone in the early to mid-1990s and associated it with *gyrA* and *parC* mutations in the quinolone-resistance-determining (QRDR) region devoted to fluoroquinolone resistance. Later, in the 2000s, this clone acquired resistance to third-generation cephalosporins due to FIB-like plasmids harbouring the *bla*_CTX-M-15_ gene and other genes conferring aminoglycoside (*strA, strB, aac(3)-IIa, aac(6’)-Ib-cr*) and quinolone (*qnrB1, oqxAB*) resistance and to other antimicrobials (*sul2, dfrA14, catB, fosA*). Finally, resistance to carbapenems emerged with several carbapenemases, including OXA-48, OXA-181, KPC-2, KPC-3, VIM-1 and NDM-1, and colistin resistance was associated with the transferable *mcr-1* gene. Nevertheless, the most urgent concern arose when the ST307 HiRiC was associated with resistance to ceftazidime-avibactam (CAZ-AVI) due to the emergence of KPC variants (KPC-2 and KPC-3, with additional mutations in the omega loop of the b-lactamase) in *K. pneumoniae* isolates. Unfortunately, there were multiple emergences in different hospitals in different geographic areas, involving outbreaks in ICUs and among patients treated with CAZ-AVI [22,29,30]. Currently, outbreaks have been reported in several countries, including Italy, Colombia, Argentina, the United States, Spain and China, and they have become endemic in some countries. This clone has a high ability to colonize hospital surfaces and also has enhanced pathogenicity [22]. In some studies, hypervirulent traits have been detected in this clone [31,32].

Other relevant recognized HiRiCs are ST383, ST392 and ST405, mostly reported as being associated with carbapenemases [33,34,35] (Table 1).

## 3. *Klebsiella pneumoniae* in ICUs: Risk Factors and Outcome

### 3.1. Updated Risk Factors

Different observational studies have attempted to determine the different risk factors associated with developing carbapenem-resistant *Klebsiella pneumoniae* (CR-Kp) infections. The population included in the main studies consisted of both adult and pediatric hospitalized patients, social-health resident patients, ICU patients, hematological patients, patients with solid-organ transplants and patients known to be CR-Kp-colonized [36,37,38,39]. The most important risk factors for CR-Kp infection reported in the published studies are prior exposure to antibiotics, ICU length of stay, use of invasive procedures and rectal colonization.

Rectal colonization is worthy of separate reflection as a risk factor for CR-Kp infection. In centres or units where culture monitoring is performed systematically, CR-Kp infection was documented almost exclusively in colonized patients [40,41,42,43]. It is interesting to ascertain the attack rate that may condition an infection. A recent study has identified that a threshold of 22% abundance for KPC-producing *Klebsiella pneumoniae* in the gastrointestinal tract is associated with the risk of developing bacteremia in hospitalized patients [32]. The sole risk factor associated with this critical threshold was prior exposure to carbapenems [43].

In addition, an Italian multicentre study analyzed the specific risk factors for developing CR-Kp bacteremia in rectal carriers [27]. Multivariate analysis revealed that admission to the ICU (OR 1.65, 95% CI 1.05–2.59, *p* = 0.03), invasive abdominal procedures (OR 1.88, 95% CI 1.16–3.04, *p* = 0.01), previous chemotherapy or radiotherapy treatments (OR 3.07, 95% CI 1.78–5.29, *p* < 0.001) and colonization of other-than-rectal colonization sites (OR 3.37 for every additional site, 95% CI 2.56–4.43, *p* < 0.001) were independent risk factors for the onset of CR-Kp bacteremia among rectal carriers. This study led to the development of the so-called “Giannella risk score” (GRS), which was significantly higher among colonized patients who developed bacteremia at 90 days compared with patients who did not develop bacteremia (8.4 vs. 2.7, *p* < 0.001). In rectal carriers with signs and symptoms of infection but not presenting the aforementioned risk factors, it would probably not be necessary to empirically use antibiotics with activity against CR-Kp [44]. The GRS has been validated in an external cohort of 94 CR-Kp-colonized patients, showing that the GRS was the best predictor of CR-Kp infection, with a cut-off of >7. It was also proposed that commencing treatment with an antibiotic with activity against CR-Kp should be considered for those rectal carriers with clinical suspicion of infection and GRS scores >7 [45].

### 3.2. Prognostic Factors

The mortality reported in the literature ranges from 10.9% and 69.3%; this broad variation is in accordance with significant differences in the populations across the different studies included in terms of severity, need for ICU admission and percentage of strains with resistance mechanisms [46,47,48].

There are some clinical circumstances associated with a worse outcome (Table 2). Predicted admission to the ICU was analyzed in 309 cases of bacteremia for *K pneumoniae*: 58 (18.8%) required ICU admission and only the pulmonary site revealed a clear association [49]. Several clinical issues have revealed a prognostic capacity for the following factors: female sex, age above 65, severity scales, use of corticosteroids, active neoplasia, severity of infection (sepsis/septic shock), respiratory site, gastrointestinal site other than the hepatobiliary system, need for mechanical ventilation and parenteral nutrition [47,49,50,51]. History of immunosuppression, high scores on prognostic scales and severity of infection are the most common factors with more overwhelming associations.

A study published in Japan found that, while bacteremia caused by *K. pneumoniae* was present in just 8.7% of strains, those strains were ESBL producers (no other resistance mechanism was detected); consequently, empirical antibiotic treatment was suitable in more than 95% of cases [47]. Interestingly, the reported mortality (in a series with 62% of patients with cancer) was just 10.9%. Conversely, regarding mortality results published in other monocentric studies with smaller numbers of patients [48], deaths of up to 52% during the first 48 h were reported. Larger studies of patients, including patients with *K. pneumoniae* bacteremia (853 cases, 178 of whom required ICU admission), showed that adequate treatment correlated with a higher survival rate in multivariate analysis [50]. In this work, the production of ESBLs was the only resistance mechanism detected (12%), and in 91.1% of cases the treatment received was inappropriate (compared to 22.1% of cases with no resistance mechanisms). Although the authors did not offer data on the distribution of suitable treatments between groups according to the existence of resistance mechanisms, Zhang et al. [52] revealed a clear association between MDR (including carbapenem resistance) and mortality, which was increased 2.942 times (95% CI 1.585–5.461; *p* < 0.001). All of the above considerations lead us to affirm the need to identify those patients with a higher probability of being carriers of MDR *K. pneumoniae* strains in order to improve the empirical treatment selection (Table 3).

Another risk factor is the existence of virulence factors. A study analyzed the impact of the presence of virulence genes by means of a multiple PCR in 129 strains retrieved from bacteremia. The presence of the iutA gene, which codes the formation of the aerobactin siderophore receptor and is therefore implicated in the virulence of the bacteria and interference with host immune systems, was significantly associated with greater mortality [47].

## 4. Clinical Entities

Although episodes of infection due to *K. pneumoniae* can also occur in the community, the bacterium is more common as an aetiological agent in nosocomial infections. Among nosocomial cases, pulmonary infections and bacteremia are especially frequent. Furthermore, *K. pneumoniae* also causes a significant number of nosocomial urinary infections. Among community-acquired entities, pneumonia, urinary infections and liver abscesses are frequent. In addition, splenic abscesses, spontaneous bacterial peritonitis, endophthalmitis and meningitis, and brain abscesses are also prevalent. The presence of some virulence factors in community-acquired infections can account for the emergence of invasive diseases due to *Klebsiella pneumoniae* [53].

### 4.1. Pneumonia

#### 4.1.1. Community-Acquired Pneumonia (CAP)

In a large study of adult patients in the US hospitalized with CAP, viruses were the main aetiological agents, whereas *Enterobacteriaceae* ranked 11th [54], with 1% prevalence. However, patients with known risk factors for Gram-negative bacteria were excluded. In a more recent article by Ceccato et al., *K. pneumoniae* was the aetiological agent in about 1% to 2% of patients with CAP [55]. Overall, *K. pneumoniae* is the aetiological agent in 1% to 7% of episodes, with 5% to 36% of these bacteria being MDR strains [56]. In Asian and African countries, the prevalence of Gram-negative microorganisms appears to be higher [57]. Unfortunately, the real incidence of *K. pneumoniae* in critical care settings is poorly described and reports are scarce. Only one study in a French ICU described 59 infections caused by *K. pneumoniae* over five years; 26 (44%) of them were community-onset infections [58].

The reported risk factors for *Klebsiella pneumoniae* CAP are female sex, diabetes mellitus and alcoholism [56]. Risk factors for MDR strains of *K. pneumoniae* are often reported together in the *Enterobacteriaceae* group. Therefore, Villafuerte et al. performed a multinational study that showed that risk factors for CAP caused by MDR strains of *Enterobacteriaceae* were prior ESBL infection, being underweight, cardiovascular diseases and hospitalization in the last 12 months [59].

Hypervirulent strains of *K. pneumoniae* can cause a syndrome of pneumonia with invasive and metastatic infections, such as bacteremia, liver abscess and necrotizing fasciitis, whereas the traditional syndrome is associated with pneumonia, urinary tract infection or infection at abdominal or surgical sites, with ulterior bacteremia [60].

#### 4.1.2. Nosocomial Pneumonia

At the beginning of the 20th century, *K. pneumoniae* represented the aetiological causative pathogen in less than 10% of nosocomial pneumonia episodes [61]. More recently, in two international and multicentre studies carried out among patients affected by nosocomial pneumonia, including ventilator-associated pneumonia (VAP) [62,63], *K. pneumoniae* was found to be the leading pathogen (37% and 33%), followed by *P. aeruginosa* (30% and 17%). The second study was focused on Gram-negative bacteria, but the first study also included Gram-positive pathogens. In the ICU setting, *K. pneumoniae* ranked second, accounting for 8.5% of isolates, as the Gram-negative aetiological agent of VAP in Spanish ICUs [2].

The clinical presentation of patients with *K. pneumoniae* nosocomial pneumonia is similar to other aetiologies. Patients can show new or persistent pulmonary opacities and fever or hypothermia, leukocytosis or leukopenia, cough, and new onset of endotracheal purulent secretions. However, bacteremia appears to be more common in nosocomial episodes than in those of community origin [53]. In the study by Juan et al., the lung was the source of bacteremia in around 6% of community-acquired episodes, compared with approximately 20% in nosocomial episodes [53].

In a meta-analysis, Agyeman et al. reported the outcomes following therapy against CR-Kp [64]. In this study, the ICU population was 12% to 100%, with most studies reporting more than 50% of ICU patients. Agyeman et al. reported that in more than 36% of the studies pneumonia represented more than 20% of CR-Kp infections [64].

#### 4.1.3. Outcome

In the study by Torres et al., focused on nosocomial pneumonia, the clinical cure rate was 76.9% in the overall population, 81.3% for patients with Gram-negative nosocomial pneumonia and 81.4% for patients with pneumonia caused by *K. pneumoniae*. All-cause mortality in the whole population was 8.4% [62]. However, in the study by Wunderink et al., comparing cefiderocol against meropenem, the mortality rate for patients with nosocomial pneumonia (including VAP) was 12% in the overall population and 10.8% when *K. pneumoniae* was the causative microorganism [63].

Patients with pneumonia caused by *K. pneumoniae*, either community-acquired or nosocomial, can have worse outcomes given certain risk factors. Hwang et al. reported that patients with bacteremia, altered mental status or respiratory rates over 22 breaths per minute presented higher mortality [60]. Mortality rates appear to be similar for traditional or hypervirulent clinical syndromes [60].

For patients with CR-Kp, the initial empirical antibiotic therapy is of major relevance to the avoidance of poor outcomes [64].

### 4.2. Bacteremia

*K. pneumoniae* is the second most common Gram-negative microorganism causing bloodstream infections in both community and nosocomial settings [2,65]. In a large study performed in Canada, with 640 episodes of *Klebsiella pneumoniae* bacteremia, an overall annual population incidence of 7.1 per 100,000 was reported [65]. According to origin, 30% of episodes were community-acquired, 43% were healthcare-associated and 27% were nosocomial. The median length of stay was 30 days in nosocomial episodes, in contrast to 7.8 and 8.1 for community or healthcare acquisition. In nosocomial bacteremia, the time from admission to diagnosis of bacteremia was 11.4 days. A source of infection was identified in 70% (*n* = 408) of episodes, the most common sources being the genitourinary tract (25%), the biliary tract (19%), intra-abdominal (10%) and pneumonia (8%).

Risk factors for *Klebsiella pneumoniae* bacteremia were solid-organ transplantation, chronic liver disease, renal dialysis and cancer. Gastrointestinal malignancies affected 13.7% of patients. Overall mortality was 20%. Mortality rates in patients with bacteremia were high but differed according to origin of acquisition: 8%, 18% and 33% in community-acquired, healthcare-associated and nosocomial episodes, respectively [65].

### 4.3. Urinary Tract Infection (UTI)

*K. pneumoniae* can cause several forms of UTI, such as cystitis, pyelonephritis, renal abscess, prostatitis and prostatic abscess. There are important differences between European countries, which have low prevalence, as compared to Asian countries. In a four-year study conducted in Croatia, Skerk et al. reported *K. pneumoniae* as the eighth most common pathogen causing chronic prostatitis [66]. However, in a shorter series in Taiwan, *Klebsiella pneumoniae* was the leading causative pathogen of prostatic abscess, causing more than 50% of cases [67]. Infections caused by ESBL-producing strains have been associated with the presence of some risk factors, such as previous urological surgery, previous hospitalization, urinary catheterization, history of renal stones and antibiotic therapy in the last three months [68,69]. Clinical manifestations of these infections are not very different from other aetiologies. In the ICU setting, UTI ranked as the seventh most common aetiology (third among Gram-negative bacteria-related aetiologies), according to recent data [2].

### 4.4. Intra-Abdominal Infections

Liver abscess due to *K. pneumoniae* is a disease that has increased in prevalence in the last few decades, especially in Asia but also in the USA [70]. The clinical picture is a set of fever, chills and right upper quadrant pain, with laboratory abnormalities, including elevated white blood cell count and alkaline phosphatase [71]. A combination of drainage and antibiotics is the standard treatment, especially in patients with larger abscesses [70].

In a large Chinese study, *Klebsiella pneumoniae* was the second most common aetiological agent in patients with spontaneous bacterial peritonitis. Initial presenting symptoms are fever, abdominal pain, vomiting and diarrhea, in variable combinations, with only fever being detected in more than half of patients with spontaneous bacterial peritonitis [72].

### 4.5. Central Nervous System (CNS) Infections

Although uncommon, *K. pneumoniae* is a frequent aetiology in CNS infections. In a Chinese retrospective study, 5.8% of the episodes mainly presented as cerebral abscess [73]. *K. pneumoniae* genotype K1 has been reported as capable of causing catastrophic septic ocular and central nervous system complications as a result of pyogenic liver abscess independent of underlying host diseases [74]. Early adequate antibiotic treatment and drainage are the key treatments for these patients.

### 4.6. The Role of Hypervirulent Klebsiella pneumoniae

Hypervirulent *K. pneumoniae* (hvKp) is an evolving pathotype that is more virulent than classical *K. pneumoniae* (cKp). hvKp usually infects individuals from the community who are otherwise healthy. Infections are more common in the Asian Pacific Rim but they occur globally. The principal risk factors associated with hvKp are ethnic background, as describe above, diabetes mellitus, male sex, immunoglobulin deficiencies, previous treatment with ampicillin or amoxicillin, and treatment of esophageal varices. Similar to cKp, hvKp strains are becoming increasingly resistant to antimicrobials via the acquisition of mobile elements carrying resistance determinants, and new hvKp strains emerge when extensively drug-resistant cKp strains acquire hvKp-specific virulence determinants, resulting in nosocomial infection [75].

hvKp infection frequently presents at multiple sites or subsequently metastatically spreads, often requiring source control. The principal sites of infection involved are pyogenic liver and splenic abscesses, community-acquired and nosocomial pneumonia (including VAP), endophthalmitis, CNS infections, necrotizing fasciitis, and genitourinary tract and endovascular infections [75].

Management challenges include rapid initiation of therapy to prevent subsequent spread, detection of occult abscesses to enable source control and appropriate site-specific therapy [75]. Although hvKp cases have higher rates of organ failure compared with non-hypervirulent cases, mortality rates are similar in the two groups [58].

## 5. Clinical Management of Infections Caused by CR-Kp

### 5.1. General Issues: Monotherapy or Combination Therapy?

The mechanism of resistance carbapenems may condition sensitivity to antibiotics and treatment decisions. There are phenotypical and genotypical tests by means of which to ascertain whether a CR-Kp produces carbapenemase and to characterize it. It is recommended that laboratories that isolate these strains have the capacity to do so or that they send them to a reference laboratory. When a strain is not sensitive to meropenem and carbapenemase cannot be studied, it is recommended to treat the infection as if it were severe [76].

There are no randomized studies to show whether combination treatment is superior to monotherapy. Evidence is based on cohort studies, mainly analyzing bacteremias [77,78,79,80]. There have been no specific studies on critically ill patients. When next-generation b-lactam antibiotics (CAZ-AVI, meropenem-vaborbactam (MER-VAB), imipenem-relebactam (IMI-REL)) are not used, combined treatment is indicated for patients at high risk of mortality (INCREMENT-CPE score > 7) (Table 4) [77,78,79,80,81,82]. When next-generation b-lactam antibiotics are used, it is not required to systematically use combined treatment. Colistin and quinolone use increases toxicity [76].

In colonized patients with infection, active empirical treatment should be used. The criteria for combined therapy use are the same. We recommend using the management algorithm derived from the ANGEL cohort [45] (Figure 1). Table 5 shows treatment options for CR-Kp infections in critically ill patients.

### 5.2. How Should CR-Kp Sensitive to Meropenem Be Treated?

Many CR-Kps that do not produce carbapenemases are resistant to ertapenem and sensitive to meropenem. This can also be the case for some strains producing carbapenemases (e.g., OXA-48). Except in the case of cystitis, most experts recommend not using meropenem as first-line treatment, especially if any carbapenemase is detected or if the agent cannot be characterized [76]. When there are no other active treatment options or none is available and when the MIC is ≤8 mg/L, high-dose meropenem could be used (2 g/8 h) in extended perfusion and combined with one or more active drugs [45,78,82,83,84,85].

KPC carbapenemases have a high affinity for ertapenem. Using ertapenem can saturate the enzyme and restore meropenem activity [86]. There is some evidence in critically ill patients for the efficacy of the ertapenem–meropenem combination in rescue treatments [87]. Whilst there are no further data, caution in using this option is recommended. Probably, it should be reserved for use as an alternative when other options are not feasible [76].

### 5.3. How Should a CR-Kp Resistant to Meropenem Be Treated?

In this situation, it is very important to identify the carbapenemase. CAZ-AVI is not active against MBL-producing strains [88]. IMI-REL and MER-VAB are not active against OXA- and MBL-producing strains [89]. Figure 2 reports the activity of each one of these drugs against the different mechanisms of resistance, and Table 5 outlines the treatment recommendations for *Klebsiella pneumoniae* infections resistant to carbapenems in critically ill patients.

In patients previously treated with CAZ-AVI, strains may appear to be resistant to this antibiotic that also appear to be sensitive to meropenem in the antibiogram; phenotypic tests may not detect carbapenemases, and if there is no sensitivity to CAZ-AVI the situation may go unnoticed [90,91,92,93].

When the strain is an OXA producer, the treatment of choice is CAZ-AVI [76,94]. Aztreonam-avibactam also has in vitro activity [95].

KPC-producing strains can be treated with CAZ-AVI [96,97,98,99], MER-VAB [100], IMI-REL [101] or cefiderocol [102]. None of these antibiotics is free of the risk of resistance [103,104,105].

CAZ-AVI has been revealed to be effective in monotherapy [98]. Nonetheless, to avoid the onset of resistant mutants, use in combined therapy in pneumonia cases is recommended when an extracorporeal blood-filtering technique is used and when there is an uncontrolled focus or a high inoculum (e.g., abdominal) [106]. Strains resistant to CAZ-AVI due to the development of mutant KPC can be treated with MER-VAB or IMI-REL [89,107]. β-lactam/β-lactamase inhibitor combinations, namely, CAZ-AVI, MER-VAB and IMI-REL are last-resort agents that represent a valuable option in the treatment of these severe infections. On the other hand, the recent emergence of resistance to these last-line antimicrobials can limit their use in clinical practice. In case of resistance to CAZ-AVI and/or MER-VAB, IMI-REL has shown adequate in vitro activity [108]. The addition of relebactam significantly improves the activity of imipenem against most species of *Enterobacteriaceae* (lowering the minimum inhibitory concentration (MIC) by 2- to 128-fold), depending on the presence or absence of β-lactamase enzymes. Against *P. aeruginosa*, the addition of relebactam also improves the activity of imipenem (MIC reduced eight-fold). Therefore, IMI-REL may be useful in patients with concomitant documented *P. aeruginosa* infections [89]. The great activity of relebactam against KPC-2 and KPC-3 β-lactamase may confer a certain advantage on the use of IMI-REL in treating these strains [109].

Colistin use is only recommended when there are no other alternatives, due to its lack of efficacy in controlled studies and high renal toxicity [76,110].

MBL-producing strains resistant to aztreonam should be treated with aztreonam-avibactam [111] or cefiderocol [112]. If aztreonam-avibactam is not available, good results have been reported using CAZ-AVI combined with aztreonam [113].

Figure 3 proposes a decision algorithm for use of the new drugs reported against CR-Kps. To sum up, CAZ-AVI may be considered for first-line treatment, and in the event of CAZ-AVI resistance and other categories mentioned in the figure, IMI-REL, MER-VAB and cefiderocol could play important roles.

## 6. Materials and Methods

The KPUCI (*Klebsiella pneumoniae* in ICU) group is a task force made up of specialists in anaesthesia, intensive care medicine, microbiology and infectious diseases, all of whom have varied experience in the field of nosocomial infections. The group carried out an extensive literature search (randomized controlled clinical trials (RCTs), systematic reviews, meta-analyses and expert consensus articles) of publications from 2012 to 2022 in the MEDLINE/PubMed and Cochrane library databases to identify relevant studies on the diagnosis and treatment of patients with *K. pneumoniae* infections and explore the main topic of the manuscript: antimicrobial therapy based on the risk of antibiotic resistance and the risk of poor outcome. Moreover, the group summarized the most important clinical entities and the antibiotic treatments that have recently been developed. After analysis of the priorities outlined, this group of experts has established a series of recommendations and designed a management algorithm.

## Figures and Tables

**Figure 1 antibiotics-11-01160-f001:**
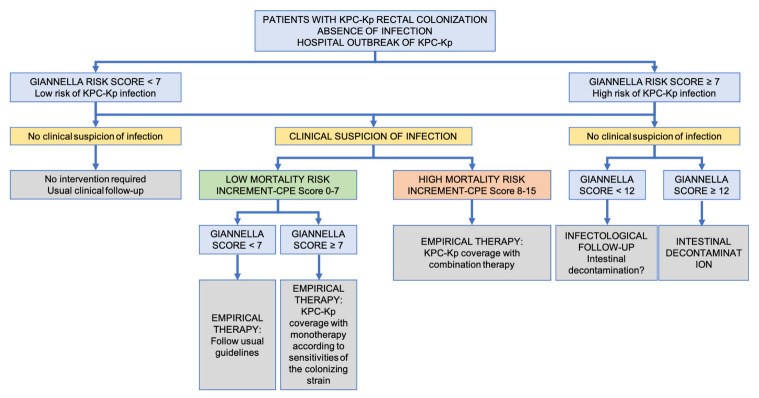
Clinical management algorithm arising from the ANGEL cohort. Reproduced with the kind permission of the authors of reference [45].

**Figure 2 antibiotics-11-01160-f002:**
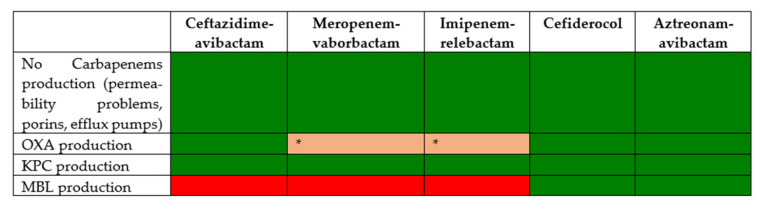
Activity of new antibiotics against *Klebsiella pneumoniae* resistant to carbapenems, according to resistance mechanisms. Green, susceptible; red, resistant. * Despite the fact that vaborbactam and relebactam are not able to inhibit OXA-48, both imipenem and meropenem have activity against some strains.

**Figure 3 antibiotics-11-01160-f003:**
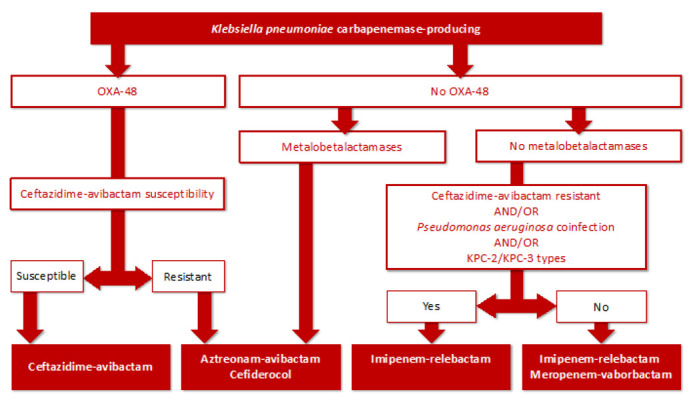
KPUCI algorithm decision to use the new drugs reported against carbapenemase-producing *K. pneumoniae*. KPUCI: *Klebsiella pneumoniae* in ICU group.

**Table 1 antibiotics-11-01160-t001:** High-risk clones in *Klebsiella pneumoniae*.

High Risk Clone	Mechanism of Antimicrobial Resistance	Co-Resistance
ST11 ST15 ST101 ST147	ESBL (CTX-M-15, …) Carbapenemases (KPC, VIM, NDM, OXA-48)	Fluoroquinolones (topoisomerases mutations, *qnr, aac(6′)-Ib-cr…)*, plasmidic AmpC beta-lactamases (DHA-1), aminoglycosides (ArmA, RmtB methylases)
ST258	Carbapenemases (KPC)	Colistin (mutations in *pmrB*), ceftazidime-avibactam
ST307	ESBL Carbapenemases (KPC-like, NDM)	Colistin (mutations in *mgrB* and *pmrB*), ceftazidime-avibactam resistance
ST383	Carbapenemases (KPC, VIM, OXA-48)	ESBL (CTX-M-15)
ST392	Carbapenemases (KPC)	Multidrug resistance
ST405	Carbapenemases (OXA-48)	Cephalosporins (ESBLs: CTX-M-14, CTX-M-15)
ST512	Carbapenemases (KPC-like)	Ceftazidime-avibactam

**Table 2 antibiotics-11-01160-t002:** Risk factors associated with mortality in *K. pneumoniae* infections.

Reference and Patients	Prediction	Prognostic Factor	OR (95% CI) *p*-Value
Namikawa et al. [47] 129 patients with bacteremia	30-day mortality	Sepsis	7.46 (1.85–30.1) <0.01
		*iutA* gen	4.47 (1.03–19.5) <0.05
Tseng et al. [49] 309 patients with community-onset bacteremia admitted to ICU	Infection-related mortality in ICU	APACHE II score	1.43 (1.12–2.01) 0.04
		Cancer	35.48 (2.54–495.57) <0.01
	1-year mortality	Cancer	3.14 (1.36–7.26) <0.01
Man et al. [50] 853 patients with bacteremia (20.9% admitted to ICU)	30-day mortality	Respiratory tract infection	2.99 (2.06–4.34) <0.01
		Intra-abdominal infection, excluding hepatobiliary	2.76 (1.76–4.34) <0.01
		Mechanical ventilation	2.20 (1.50–3.22) <0.01
		Medical ward	1.83 (1.25–2.67) <0.01
		No diabetes mellitus	1.76 (1.24–2.50) <0.01
		Inappropriate empirical antibiotic treatment	1.71 (1.27–2.32) <0.01
		Female sex	1.70 (1.25–2.31) <0.01
		Age > 65 years	1.69 (1.16–2.47) <0.01
		Solid tumour	1.46 (1.05–2.01) 0.02
Papadimitriou-Olivgeris et al. [51] 139 patients with ICU-acquired bacteremia (CP-Kp)	30-day mortality	Septic shock	6.5 (2.2–19.5) <0.01
		SAPS II	1.1 (1.0–1.2) <0.01
		Corticosteroids during bacteremia treatment	3.1 (1.1–8.6) 0.02
		Parenteral nutrition	2.8 (1.0–7.7) 0.04
		Combination antibiotic treatment	0.24 (0.07–0.75) 0.01

APACHE II: Acute Physiology and Chronic Health Evaluation II, CP-Kp: carbapenemase-producing *K. pneumoniae*, ICU: intensive care unit, SAPS II: Simplified Acute Physiology Score II.

**Table 3 antibiotics-11-01160-t003:** Risk factors associated with the existence of resistance mechanisms in *K. pneumoniae* infections.

References		OR (95% CI)	*p*-Value
**Extended-spectrum beta-lactamases**			
Man et al. [50]	No hepatobiliary site	2.231 (1.341–3.712)	0.002
	Corticosteroids in the previous 30 days	1.957 (1.061–3.610)	0.032
	Solid cancer	1.851 (1.214–2.823)	0.004
	CVC carrier in the current admission	1.686 (1.051–2.705)	0.030
**Resistance to carbapenems**			
	Prior admission to ICU	2.32 (1.22–4.4)	0.01
Zhang et al. [52]	Surgery	2.33 (1.26–4.32)	0.007
	Prior antibiotics	2.02 (1.1–3.74)	0.024
	Mechanical ventilation	3.3 (1.56–6.97)	0.002
	APACHE II > 10 points	1.9 (1.06–3.42)	0.031
	Hospital stay > 14 days	4.34 (2.21–8.55)	<0.001

CVC: central venous catheter.

**Table 4 antibiotics-11-01160-t004:** INCREMENT-CPE score.

Population	Variables	Score
General ^a^	Severe sepsis or septic shock	5
	Pitt score ≥ 6	4
	Charlson comorbidity index	3
	Source other than urinary or biliary tract	3
	Inappropriate early targeted therapy	2
	Maximum score	17
Solid-organ transplant ^b^	INCREMENT-CPE score ≥ 8	8
	Cytomegalovirus disease (previous 3 days)	7
	Lymphocytes ≤ 600 mm^3^	4
	No source control	3
	Inappropriate empirical therapy	2
	Interaction INCREMENT-CPE score ≥ 8 * Cytomegalovirus disease (previous 3 days)	–7
	**Maximum score ^c^**	**17**

^a^ See reference [81]. ^b^ See reference [82]. ^c^ The maximum score in a patient with all risk factors would be 17 (INCREMENT-CPE score ≥ 8 (+8), cytomegalovirus disease (+7), lymphopoenia (+4), no source control (+3), inappropriate empirical therapy (+2) and interaction INCREMENT-CPE score ≥ 8 with CMV (−7)).

**Table 5 antibiotics-11-01160-t005:** Treatment recommendations for CR-Kp infections in critically ill patients.

Carbapenem-Resistant *Klebsiella pneumoniae*	Recommended Treatment	Alternative Therapy
**No carbapenemase production or not available**		
Resistant to ertapenem and susceptible to meropenem	Meropenem (extended infusion)	Ceftazidime-avibactam Quinolones, trimethropim-sulfamethoxazole, nitrofurantoin, aminoglycosides can be used in cystitis if susceptible
Resistant to Ertapenem and meropenem	Ceftazidime-avibactam, Meropenem-vaborbactam, Imipenem-relebactam, Cefiderocol	Aztreonam-avibactam ^b^ Tigecycline (in combination therapy), Eravacycline (only with abdominal source) Quinolones, trimethropim-sulfamethoxazole, nitrofurantoin, aminoglycoside can be used in cystitis if susceptible
**Carbapenemase production**		
Oxacillinase carbapenemases (OXAs)	Ceftazidime-avibactam Cefiderocol	Aztreonam-avibactam ^b^ Tigecycline (in combination therapy), Eravacycline (only with abdominal source), Aminoglycoside (urinary tract infections, catheter-related infections, combined therapy (including Plazomycin), Fosfomycin (urinary tract infections)
Klebsiella pneumoniae carbapenemases (KPCs)	Ceftazidime-avibactam, Imipenem-relebactam, Meropenem-vaborbactam, Cefiderocol	Aztreonam-avibactam ^b^ Tigecycline (in combination therapy), Eravacycline (only with abdominal source) Amynoglicoside (urinary tract infections, catheter-related infections), combined therapy (including Plazomycin), Fosfomycin (urinary tract infections)
Metallo-b-lactamases ^a^ (MBLs)	Aztreonam (when susceptible) Aztreonam-avibactam ^b^	Tigecycline (in combination therapy), Eravacycline (only with abdominal source) Amynoglicoside (urinary tract infections, catheter-related infections, combined therapy (including Plazomycin), Fosfomycin (urinary tract infections)

^a^ Verona integron-encoded metallo-β-lactamases (VIMs), New Delhi metallo-β-lactamases (NDMs) or imipenem-hydrolyzing metallo-β-lactamases (IMPs). ^b^ Ceftazidime-avibactam plus aztreonam can be used until available.

## Data Availability

Not applicable.

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
