# Peer review of "Current Positioning against Severe Infections Due to Klebsiella pneumoniae in Hospitalized Adults"

_antibiotics, 2022, doi:10.3390/antibiotics11091160_

Round 1

Reviewer 1 Report

Congratulations to the Authors per this manuscript and for their knowledge of the topic. Furthermore, the antibiotic therapy is always an hot topic, especially in critical ill setting, such as ICU or Emergency Department, where de facto the antibiotic therapy starts.

Because of the many variables and risk factors that can influence the diagnosis, therapy and prognosis, it would be desirable that, alongside the indispensable international guidelines, there were personalized antibiotic therapy guidelines for countries or specific geographical areas.

Author Response

Congratulations to the Authors per this manuscript and for their knowledge of the topic. Furthermore, the antibiotic therapy is always a hot topic, especially in critical ill setting, such as ICU or Emergency Department, where de facto the antibiotic therapy starts.

Because of the many variables and risk factors that can influence the diagnosis, therapy and prognosis, it would be desirable that, alongside the indispensable international guidelines, there were personalized antibiotic therapy guidelines for countries or specific geographical areas.

Response 1.

Thank you very much for your comment. We fully agree: international guidelines are important, but they should be adapted to local microbiology. In our manuscript we try to show which is the best therapy in each microbiological situation so that, with this information, each unit can develop its own protocols.

Reviewer 2 Report

First and foremost, I would like to commend the authors for this review that address such an important issue that could impact patient outcomes. The following comments are worth attention before considering this manuscript for publication:

·      Lines 269 – 270: The sentence was read in a complex way. I advise the authors to rewrite it more straightforwardly.

·  Lines 280 – 287: This paragraph describes the epidemiology of community-acquired pneumonia caused by K. pneumoniae among hospitalized patients. However, it does not specify if those numbers pertain to ICU versus non-ICU patients. This distinction is essential since this paper focuses on ICU patients.

·   Lines 374 – 375: The authors mentioned that K. pneumoniae is the second most common etiological agent of SBP. However, the reference provided for this statement is a study in China. It is recommended to include more references from other countries to support the global application of this statement or specifically mention that this prevalence was reported in China.

Author Response

Response to Reviewer 2 Comments

First and foremost, I would like to commend the authors for this review that address such an important issue that could impact patient outcomes.

Thank you very much for your comment.

The following comments are worth attention before considering this manuscript for publication:

  • Lines 269 – 270: The sentence was read in a complex way. I advise the authors to rewrite it more straightforwardly.

We agree. We have rewritten the sentence

  • Lines 280 – 287: This paragraph describes the epidemiology of community-acquired pneumonia caused by K. pneumoniae among hospitalized patients. However, it does not specify if those numbers pertain to ICU versus non-ICU patients. This distinction is essential since this paper focuses on ICU patients.

Thank you for the comment. It is true. Unfortunately, the real incidence ok K. pneumoniae in critical care setting is poorly described and scarse, only one study in a French ICU [Rafat C, Messika J, Barnaud G, Dufour N, Magdoud F, Billard-Pomarès T, et al. Hypervirulent Klebsiella pneumoniae, a 5-year study in a French ICU. J Med Microbiol. 2018 Aug;67(8):1083–9] described 59 infections caused by K. pneumoniae during five years; 26 (44%) of them were community-onset infections. A comment has been added in the paragraph.

Lines 374 – 375: The authors mentioned that K. pneumoniae is the second most common etiological agent of SBP. However, the reference provided for this statement is a study in China. It is recommended to include more references from other countries to support the global application of this statement or specifically mention that this prevalence was reported in China.

We have specifically mentioned in the text that this prevalence has been reported in China.

We sincerely appreciate your comments, which have contributed to improve the quality of the manuscript.

Reviewer 3 Report

In this review ‘Current Positioning against K. pneumoniae in the ICU,’ Vidal-Cortes et al aim to discuss the current epidemiology, pathogeny, risk factors for poor outcomes, clinical syndromes, and management of infections caused by Klebsiella pneumoniae, particularly shedding light on algorithm-based treatment options of carbapenem-resistant Enterobacterales infections, based on published literature over the past decade.  

Overall, the review is well-summarized. One of the biggest strengths of this study is it is backed up with strong data and flows well. While this paper has certainly been engaging, there are some issues in this paper that require explanation. Please see comments below for specific concerns.

Major comments

-Unclear why the title and other subheadings say in the ‘ICU’, when most of the studies included in this review were done in different healthcare settings and not just limited to ICU. The title can be modified to ‘hospitalized adults.’

-Since this review focuses on management of Klebsiella pneumoniae, hypervirulent Klebsiella is a subtopic that deserves more attention, given the severe pathogenicity associated with it. It is only mentioned in lines 325-329, but the review would benefit from further elaboration of hypervirulent Klebsiella in different sections such as epidemiology, risk factors, clinical entities, and management. After line 377, can consider adding brain abscess and other severe forms of infection this can cause.

-Certain figures and tables are directly taken from the reference papers as it is (screenshot), would prefer authors rewrite those figures and tables.

-The paper will benefit from further proof-reading due to use of incorrect grammar and sentence formation, some of which are outlined below.

Minor comments

Abstract – Active voice is usually preferred over passive voice, writing style can be improvised.

1. Introduction

Line 60 – 62: The first two sentences in introduction are exactly same as abstract, can consider modifying the language.

2. Current epidemiology, pathogeny and antimicrobial resistance of Kp infections in the ICU

Line 71:  The link to reference 1 does not open, please provide the correct citation. If this citation is based on Spanish data only, please add ‘in Spain’ at the end of the sentence.

Line 88: replace ‘K. pneumoniae carbapenemase, KPC as well as BKC and SME’ with ‘K. pneumoniae carbapenemase (KPC), BKC and SME’

Line 124: replace ‘being MBLs and KPC enzymes more efficient’ with ‘with MBLs and KPC enzymes being more efficient’

Line 133: rephrase ‘and invisible for diagnostic detection systems’ to make the phrase clearer

3. Kp in ICU. Risk factors and outcome

Line 223: presented ‘with’ GRS ≥ 7.

Line 226: please explain ‘the series’ – elaborate on the sites and nature of these studies conducted. You can alternatively state ‘The mortality reported in the literature varies ranges from 10.9% and 69.3%; this broad variation is in accordance with significant differences in the population across different studies included such as severity……. ‘

Lines 233-239: it would be helpful to add OR to understand how strong these associations were. Also explain ‘prognostic capacity’ - did it mean mortality vs. severity of illness? Which factors were associated with which outcomes? Alternatively, add these studies to table 2 and elaborate on the outcomes studied in the table.

Line 254: ‘revealed’

4. Clinical entities

Line 269: remove ‘by’, and say ‘both in the community and the hospital setting’

Line 280: rephrase to say ‘hospitalized with CAP’

Line 281: remove ‘of’

Line 308: add 6% of ‘community-acquired’ episodes

Line 314: replace CRKP with CR-Kp

Lines 316 – 318: please rewrite to ensure clarity as the current verbiage is not clear

Line 346: rephrase ‘being the most common sources’ with ‘the most common sources being’

Line 356, mention where the Skerk et al study was based at, for example – In a four-year study based at ***, Skerk  et al reported…..’

5. Clinical management of infections caused by CR-Kp

Line 427: please rephrase as ‘activity of these drugs against’

Line 442: please replace ‘has also’ with ‘also has’

Line 447: specify combined therapy with which antibiotics?

Lines 458-460: since this topic is focused on Enterobacteriaceae, not sure if adding treatment options for resistant Pseudomonas infections is necessary, and would suggest deleting this as management of resistant Pseudomonas is a separate topic on its own.

6. Materials and Methods

-Active voice is usually preferred over passive voice

-The writing style can be improvised

-Language can be made slightly different from the abstract

-Would elaborate further how these studies were chosen

7. Figures/Tables

Figure 1: would urge the authors to reproduce it rather than copying a screenshot exactly from the reference. If you decide to keep as is, would ensure the screenshot is not cut (box on the left side is cut) and the asterisk (*) are denoted in the footnote

Table 4: Footnote/superscript ‘a’ has been incorrectly placed. It should go under MBLs category and not under KPCs – VIMs, NDMs and IMPs are MBLs and not KPCs.

Figure 3: Please specify what KPICI stands for?

Also for ‘No OXA-48, No Metalobetalactamases’ category, it would be helpful to list somewhere that CAZ-AVI is the first-line, and in event of CAZ-AVI resistance and other categories mentioned, can go to IMI-REL or MER-VAB as listed.

Round 2

Reviewer 3 Report

The manuscript has undergone significant revisions and has improved. I believe it can be accepted for publication in its present form.